Back pain in the midwifery profession in northern Poland

Bryndal Aleksandra aleksandra.bryndal@upsl.edu.pl 1 2
Glowinski Sebastian sebastian.glowinski@upsl.edu.pl sebastian.glowinski@tu.koszalin.pl 1 2
Hebel Kazimiera 1
Grochulska Agnieszka 1
1 Institute of Health Sciences, Pomeranian University in Slupsk , Slupsk , Poland
2 State Higher School of Vocational Education in Koszalin , Koszalin , Poland
Jimenez-Cebrian Ana Maria
Electronic publication date: 2025 Mar 26
Publication date: 2025
Volume: 13
Electronic Location ID: e19079
Received 2024 Oct 18; Accepted 2025 Feb 10
Copyright: ©2025 Bryndal et al.
Copyright year: 2025
Copyright holder: Bryndal et al.
License: This is an open access article distributed under the terms of the Creative Commons Attribution License, which permits unrestricted use, distribution, reproduction and adaptation in any medium and for any purpose provided that it is properly attributed. For attribution, the original author(s), title, publication source (PeerJ) and either DOI or URL of the article must be cited.
License URL: https://creativecommons.org/licenses/by/4.0/

Keywords: Midwife, Neck pain, Low back pain, NDI, ODI, Medical staff

Funding: The authors received no funding for this work.

==============================
Background

Neck pain (NP) and low back pain (LBP) are increasingly significant medical, social, and economic concerns. The midwifery profession, similar to other healthcare occupations, is particularly predisposed to these issues.

Methods

This study aimed to analyze the prevalence of back pain among midwives and evaluate the associated disability levels using the Neck Disability Index (NDI) and the Oswestry Disability Index (ODI). The study group included 208 actively practicing midwives aged 23 to 67 years (mean ± SD: 48.1 ± 10.7 years). Participants completed an anonymous survey comprising a custom-designed questionnaire, the Polish language versions of the NDI and ODI, and the Visual Analog Scale (VAS) for pain intensity.

Results

The analysis revealed a statistically significant correlation between VAS pain intensity and both age (r = 0.2476) and work experience (r = 0.2758), indicating higher pain scores with increasing age and seniority. No significant association was found between BMI and VAS scores (r = 0.0011). Additionally, NDI and ODI scores correlated significantly with age (r = 0.1731; r = 0.3338), BMI (r = 0.1685; r = 0.2718), and work experience (r = 0.1987; r = 0.4074). Higher values for age, BMI, and seniority were associated with increased disability levels.

Conclusions

Neck and low back pain represent prevalent and impactful issues for midwives in Poland, contributing to mild to moderate disability, absenteeism, reliance on pain medication, and limited physical activity. Key contributing factors include age, professional experience, BMI, and low levels of physical activity.

Introduction

According to the European Working Conditions Survey (EWCS), back complaints are the most commonly reported work-related health problems in the European Union (Eurofound et al., 2017). Neck pain (NP) and low back pain (LBP) are ailments that are becoming a growing medical, social and economic problem (Saragiotto et al., 2016; Shemshaki et al., 2013; Kassolik et al., 2017). NP is ranked as the fourth cause of disability (Murray et al., 2013). In the adult population between 15 and 74 years of age, the prevalence of NP is defined as 5.9% to 38.7% (Cote, Cassidy & Carroll, 1998; Popescu & Lee, 2020). NP is more common in women than men (Cohen & Hooten, 2017; Bryndal et al., 2023; Bryndal, Glowinski & Grochulska, 2022; Glowinski, Bryndal & Grochulska, 2021). Inadequate ergonomics at work, sitting and keeping the neck posture in an unphysiological position for long periods of time are identified as the main causes of NP (Bryndal et al., 2023; Bryndal, Glowinski & Grochulska, 2022; Glowinski, Bryndal & Grochulska, 2021; Popescu & Lee, 2020).

The prevalence of LBP in the adult patient population is up to 84% over a lifetime (Bryndal et al., 2023; Bryndal, Glowinski & Grochulska, 2022; Glowinski, Bryndal & Grochulska, 2021; Popescu & Lee, 2020; Deyo & Tsui-Wu, 1987), and of chronic back pain about 37% with often accompanied by functional limitations and reduced performance (Fleckenstein et al., 2022). The Global Burden of Diseases, Injuries, and Risk Factors Study (GBD) on the prevalence of back pain (Wu et al., 2020; Hoy et al., 2014; Chen et al., 2021) confirmed that LBP is a major cause of disability in most countries. Both the total disability burden and disease-related costs are expected to continue to rise in the coming decades (Hartvigsen et al., 2018). The incidence of disability due to LBP increased with age, with a peak in prevalence rates observed at around 85 years of age. Global prevalence rates were higher among women compared to men in all age groups, although more pronounced differences were observed in older age groups (i.e., over 75 years). The global age-standardised prevalence per 100 000 was also higher in women (9330; 95% UI 8370-10 500) compared with men (5520; 4930-6190) (GBD, 2023).

The prevalence of activities that are limited by LBP and significantly impede work and quality of life for at least 1 day is determined to be 12%. The incidence of LBP within one month is determined to be 20–26% (Hoy et al., 2012). In patients who experience acute LBP and present to a doctor, the complaints resolve in 70–90% of cases (Popescu & Lee, 2020; George et al., 2021). However, even if acute LBP subsides, up to 70% of patients, may experience a repeat episode of pain within a year, while 54% may experience a recurrence within six months. The occurrence of LBP predisposes to another episode (Pengel et al., 2003; Mehling et al., 2012). LBP is identified as a leading cause of physical disability and loss of productivity worldwide (Rubin, 2007; Saragiotto et al., 2016; Fleckenstein et al., 2022; De Campos, 2017; Vos et al., 2012). LBP can be classified as an occupational disease, in terms of exposure to bending the trunk or lifting and carrying heavy objects (Kuijer et al., 2018). The Sixth European Working Conditions Survey (SEWCS) report shows that those with the highest work intensity are those who provide healthcare, including physiotherapists, nurses, midwives and paramedics (Eurofound et al., 2017). It also results in high exposure to trauma, prolonged static and/or dynamic overload of the musculoskeletal system and back pain (Bryndal et al., 2023; Bryndal, Glowinski & Grochulska, 2022; Glowinski, Bryndal & Grochulska, 2021; Mroczek et al., 2020). The type of activities performed in the medical professions has an impact on the spine, as working with patients involves forced, prolonged maintenance of an often unnatural body position. The occupational activities of medical staff are often repeated, performed in a standing position with the trunk bent or twisted and involve lifting, especially during nursing activities and medical procedures. Lifting mothers out of bed (to help them sit up or walk) and breastfeeding training are cited as the activities that put the most strain on the musculoskeletal system in midwifery work (Hasheminejad, Amirmahani & Tahernejad, 2023). The nature of midwifery work generates the adoption of body positions in which the spinal alignment considered most unfavourable—flexion combined with rotation—occurred. These movements are, in a way, part of the professional activity in the form of forced positions (Report of the Supreme Council of Nurses and Midwives, 2017; Bryndal et al., 2023; Bryndal, Glowinski & Grochulska, 2022; Glowinski, Bryndal & Grochulska, 2021). According to the Sixth European Working Conditions Survey (SEWCS), the cervical and lumbar spine, upper limbs and feet are considered the most vulnerable areas in midwives (Eurofound et al., 2017; Hasheminejad, Amirmahani & Tahernejad, 2023). Another important factor that further contributes to the occurrence of psycho-physiological problems, including LBP, is the extended working hours of midwives in relation to the statutory hours and, consequently, the lack of rest (Matsudaira et al., 2013; Hochhauser & Liberman, 2024). However, common to the reports cited above and to our own findings is the frequent occurrence of back pain, which may be attributed to occupational activities performed in positions that deviate significantly from ergonomic standards. This presents a challenge for healthcare systems, necessitating new strategies for managing workplace ergonomics and preventing occupational injuries. Clinical research highlights a lack of knowledge and awareness regarding ergonomics, as well as difficulties in applying this knowledge in clinical practice within medical professions (Bernardes et al., 2022; Hochhauser & Liberman, 2024).

Aim

The aim of this study was to analyse the prevalence of back pain in midwives and to assess the rate of disability assessed by the NDI and ODI resulting from these complaints.

Materials & Methods

An epidemiological and cross-sectional study was conducted to collect data on the prevalence of back pain among midwives. This study was approved by the Bioethics Committee at the District Medical Chambers in Gdansk (KB-14/20). All participants were made aware of the purpose of the study and gave informed written consent to participate in the study.

Population

The study group consisted of 208 women who were active midwives and anonymously completed a questionnaire on complaints of back pain and the nature of their work. Data was collected between January 2023 and January 2024.

Selection criteria

Eligible for the study were those over 18 years of age, active and licensed as midwives. The following were used as exclusion criteria: age under 18 years, history of spinal injury and/or surgery, diseases of the nervous system, spinal and/or limb deformities and pregnant women were excluded from the study. The reason for this was the static changes in the musculoskeletal system during pregnancy and the resulting possible occurrence of spinal pain.

Measurement instrument

The research used a questionnaire consisting of three parts. The first part was the author’s “self-administered online questionnaire”. The second part is the Neck Disability Index (NDI) questionnaire Polish language version (Misterska, Jankowski & Glowacki, 2011a). While the third part is the Revised Oswestry Disability Index (ODI) Polish language version (Misterska, Jankowski & Glowacki, 2011b).

The author’s “self-administered online questionnaire” included questions on demographic data such as age, weight and height values. From these data, body mass index (BMI) was calculated. BMI was calculated as weight (kg) divided by height squared (m2) (Bryndal et al., 2023; Bryndal, Glowinski & Grochulska, 2022; Glowinski, Bryndal & Grochulska, 2021). Questions were also asked about the characteristics of the job, where they asked about the type of activities dominating the job, the number of working hours per month and the length of service. They were asked about the prevalence of current and past complaints of back pain, such as the location of the pain, the duration of the pain, the continuity of the symptoms, the presumed cause or causes and the positions that aggravate the pain. Back pain was defined as continuous or intermittent pain lasting 3 or more days, and respondents were asked to mark the appropriate response indicating the location of the pain. The physical activity performed was also asked about physical activity as broadly defined as physical recreation (Bryndal et al., 2023; Bryndal, Glowinski & Grochulska, 2022; Glowinski, Bryndal & Grochulska, 2021).

In those who experienced pain, the intensity of the pain was determined using the Visual Analogue Scale (VAS). The VAS scale took the form of a ten-centimetre ruler on which the respondent marked the severity of pain on a scale from 0, which was defined as no pain at all, to 10, which was defined as the most severe pain imaginable (Bryndal et al., 2023; Bryndal, Glowinski & Grochulska, 2022; Glowinski, Bryndal & Grochulska, 2021).

The Neck Disability Index (NDI) Polish language version questionnaire was used to determine the rate of cervical disability due to NP (Misterska, Jankowski & Glowacki, 2011a). It consists of 10 questions on pain intensity, grooming, lifting objects, reading, headaches, ability to focus, working, driving, sleeping, resting.

The Oswestry Low Back Pain Disability Scale (ODI) Polish language version (Misterska, Jankowski & Glowacki, 2011b) questionnaire was used to determine the disability rate due to LBP. The authors have permission to use this instrument from the copyright holders. It consists of 10 questions on pain intensity, grooming, lifting, walking, sitting, standing, sleeping, sex life, social life, travelling. On the NDI and ODI scales, each question has six possible answers rated on a point scale from 0 to 5. The final score separately for each questionnaire is presented on a point scale from 0 to 50 or a percentage from 0 to 100%. The score is classified into an appropriate point or percentage range that determines the level of disability: 0–4 pts. (0–8%) no disability, 5–14 points. (10–28%) mild disability, 15–24 points (30–48%) moderate disability, 25–34 points (50–64%) severe disability, 35–50 points (70–100%) extreme suffering and disability (Misterska, Jankowski & Glowacki, 2011a; Misterska, Jankowski & Glowacki, 2011b).

Test procedure

The author’s NDI and ODI “self-administered online questionnaire” was distributed electronically to respondents to the email addresses of medical facilities and departments employing midwives (Misterska, Jankowski & Glowacki, 2011a; Misterska, Jankowski & Glowacki, 2011b). The questionnaire was sent electronically to medical facilities located in northern Poland in the Pomeranian Voivodeship that employ midwives, based on the list of the Pomeranian Voivodeship Branch of the National Health Fund. In the Pomeranian Voivodeship in 2023, the occupational activity rate was 63.3% (1,303 midwives) out of the registered 2,058 midwives. Nationwide in Poland, the occupational activity rate was 70% (29,203 midwives) out of the registered 41,719 midwives (MedMedia, 2023). At the beginning of the questionnaire, an information form about the study and informed consent was presented. We informed the respondent of the anonymity of the survey and that the results of the survey would be used in scientific research and suggested that the responses should be honest. 241 people agreed to take part in the study, of whom eight did not meet the inclusion criteria and the remaining 25 did not complete the questionnaire correctly.

Statistical analysis

All statistical analyses were performed using the STATISTICA software, version 13.3 (2020), from StatSoft Inc. (Palo Alto, CA, USA). Descriptive statistics for quantitative variables included the arithmetic mean, standard deviation, median, minimum and maximum values (range), and 95% confidence intervals (CI) (Glowinski, Bryndal & Grochulska, 2021). The Shapiro–Wilk W test, Lilliefors test, Kolmogorov–Smirnov test, and Jarque–Bera test were applied to assess whether a quantitative variable followed a normal distribution. The Levene (Brown-Forsythe) test was employed to examine the homogeneity of variances. For comparing differences between two groups (independent samples), the Student’s t-test was used when variances were equal, and the Mann–Whitney U test was used when the assumptions for the t-test were not met. The Kruskal–Wallis test was applied to evaluate differences within the same variable across different groups when the normality assumption was violated, followed by a post hoc test if a significant result was found. Correlation analysis used Pearson’s correlation coefficients to explore the strength and direction of relationships between variables. Before investigating these relationships, scatter plots were generated to assess the potential presence of outliers visually. A significance level of p = 0.05 was used for all tests, with significant results highlighted in bold in the tables.

Results

The study group consisted of 208 female midwives with an age range of 23 to 67 years (mean (SD) 48.1 (10.7)), a height of 153 to 179 cm (mean (SD) 165.3 (6.0)), a weight of 44 to 99 kg (mean (SD) 68.9 (11.0)) and a BMI of 17.6 to 37.7 (mean (SD) 25.3 (4.2)). By age category, 17 (8.2%) were in the 20–29 age bracket, 32 (15.4%) in the 30–39 age bracket, 42 (20.2%) in the 40–49 age bracket, while 117 (56.2%) were in the ≥ 50 age bracket.

Job characteristics of the study group of midwives

Length of service in the study group ranged from 1 to 45 years of service (mean (SD) 24.4 (11.6)). The number of working hours per month ranged from 48 to 280 h (mean (SD) 167.6 (26.6)). The working system as single shift was declared by 34 midwives (16.3%), double shift by 146 midwives (70.2%), 24-hour on-call by 3 midwives (1.4%) and mixed system by 25 midwives (12.0%).

Characteristics of pain in the study group of midwives

Back pain during their working lives was declared by 203 midwives (97.3%). In the back pain group, 116 women (57.1%) declared NP of which 68 (58.6%) had pain centrally, 23 (19.8%) had pain with radiation to one upper limb, while 25 (21.6%) had pain with radiation to both upper limbs. In the group of people with back pain, 41 (19.7%) declared pain centrally in the thoracic part of the spine. In the group of midwives with back pain, as many as 192 (94.6%) declared LBP, of whom 96 (50.0%) had pain centrally, 87 (45.3%) had pain radiating to one lower limb, while nine (4.7%) had pain radiating to both lower limbs. Respondents in the question that dealt with the occurrence of pain at a particular level of the spine were given the opportunity to select several answers, which resulted in the number of respondents in a given group and the percentage distribution may be more than 100%.

The intensity of pain on the VAS scale was rated from 2 to 10 (mean (SD) 4.65 (1.53)). Respondents rated their pain on the VAS scale most frequently as 3 (mild/moderate) 22 people (10.68%), 4 (moderate) 81 people (39.32%), 5 (moderate/strong) 42 people (20.39%) and 6 (strong) 29 people (14.08%).

The time of the first incident of back pain was described by 10 people (4.85%) as one year ago, 44 (21.36%) as 2–3 years ago, 55 (26.70%) as 4–6 years ago, 18 (8.74%) as 7–9 years ago, 79 (38.35%) as more than 10 years ago, while 2 people (0.96%) did not answer this question. The number of spinal pain episodes in the past was defined as zero by seven people (3.37%), between 1–5 was defined by 80 people (38.46%), between 6–10 was defined by 33 people (15.87%) while 11 or more episodes was defined by 88 people (42.31%).

The nature of the pain and its frequency in the study group was rated by respondents as constant pain with 47 people (22.82%). Pain that comes and goes once a day–35 (16.99%) once a week–51 (24.96%), once a month–28 (13.59%), once a year–9 (4.37%). Pain that appeared and subsided several times in their working lives was declared by 33 midwives (16.02%), while one that appeared and subsided only once in their working lives was declared by three midwives (1.46%).

In the group of midwives with back pain, 69 women (34.0%) declared the use of sick leave due to these complaints, while the use of painkillers for the same reason was declared by 144 women (70.9%). In the question on the use of treatment for back pain other than painkillers, 33 people (16.02%) declared no treatment, 23 people (11.17%) used medical advice, 69 people (33.50%) used rehabilitation/physiotherapy treatment privately, while 60 people (29.13%) used insurance, and other treatments were declared by 21 people (10.19%).

Respondents were also asked whether they had any mobility limitations due to back pain 206 of the total sample answered. In response to this question, 47 people (22.82%) said they had no limitations, 133 (64.56%) had partial limitations, 22 (10.68%) had limitations that made it difficult to work and three people (1.46%) said limitations prevented them from functioning independently.

With regard to the maintenance of ergonomics at work, midwives were asked whether they use aids when performing physically demanding tasks, with 205 of the entire study group responding. One hundred women (48.54%) are assisted by a second person, zero (0.00%) are assisted by a hoist, 100 (48.54%) perform these activities alone, while five people (2.43%) use another method.

Of the 208 midwives participating in the survey, 206 responded to the question on identifying the activities that generate the occurrence of pain, 108 (52.43%) identified that after lifting a load, 128 (62.14%) that during bending, 78 (37.86%) during standing, 65 (31.55%) during sitting, 68 (33.01%) during trunk twisting, 21 (10.19%) during trunk hyperextension. Respondents could have given several answers to this question, which is why the total % is more than 100%.

Physical activity in the study group of midwives

The nature of the physical activity undertaken was determined in the study group, with 205 midwives responding to these questions. In defining the type of physical activity undertaken, 18 (8.74%) declared that they did not undertake physical activity, 83 (40.29%) described it as mild physical activity, 101 (49.03%) described it as moderate, medium physical activity, while three (1.46%) described it as intense physical activity. The duration of a single physical activity was described by 19 people (9.22%) as none, 87 people (42.23%) declared 10–30 min, 64 people (31.07%) declared 30–50 min, while 35 people (16.99%) declared 50 min or more. The frequency of physical activity per week was described by 40 people (19.42%) as one time or zero, 102 people (49.51%) described it as 2–3 times, 40 people (19.42%) described it as 4–5 times, while 17 (8.25%) described it as 6–7 times.

ODI and NDI results among midwives

In the entire study group, 116 women declared NP, while 206 midwives completed the NDI questionnaire. Among those surveyed, 192 women declared LBP, but 206 midwives completed the ODI questionnaire. Table 1 presents the characteristics of participants with LBP who completed the ODI and NDI questionnaires (n = 206 for both scales). The mean ODI score was 7.62 (SD = 4.99, 95% CI [6.94–8.31], median = 7, range 0–25), while the mean NDI score was 7.74 (SD = 6.11, 95% CI [6.90–8.58], median = 7, range 0–23). Most participants reported minimal disability, with 60 (61.65%) in the “no disability” category (0–4 points) on the ODI and 66 (32.04%) on the NDI. Light disability (5–14 points) was the most prevalent category for both scales, accounting for 127 (61.65%) and 109 (52.91%) participants, respectively. Mild disability (15–24 points) affected 17 (8.25%) on the ODI and 31 (15.05%) on the NDI, while severe disability (25–34 points) was rare (ODI: 2 participants, 0.97%; NDI: 0 participants). No participants reported extreme suffering or disability (35–50 points) on either scale. This data highlights the generally low disability levels in this cohort, with light and mild disability being the most common.

Table 1 ODI and NDI results among midwives.

Number of participants with LBP who completed the questionnaire	 	 	ODI (n = 206)	NDI (n = 206)	
Mean (SD) [95% CI] Me Range		 	7.62 (4.99) [6.94–8.31] 7 0–25	7.74 (6.11) [6.90–8.58] 7 0–23	
0–4 pts (0–8%) no disability	Number (% of the group with NP)	 	60 (61.65%)	66 (32.04%)	
5–14 pts (10–28%) light disability	 	127 (61.65%)	109 (52.91%)	
15–24 points (30–48%) mild disability	 	17 (8.25%)	31 (15.05%)	
25–34 pts (50–64%) severe disability	 	2 (0.97%)	0 (0.00%)	
35–50 pts (70–100%) extreme suffering and disability	 	0 (0.00%)	0 (0.00%)	
Notes.

ODI Oswestry Low Back Pain Disability Scale

NDI Neck Disability Index

LBP Low Back Pain

NP Neck Pain

Level of significance for selected relationships

In a study group of midwives reporting back pain during their professional careers, the intensity of pain, as measured by the VAS scale, showed a statistically significant correlation with both age (R = 0.2476) and work experience (R = 0.2758). The results indicated that higher age and longer work experience were associated with higher pain scores. Each data point represents an individual, and the trend line illustrates a slight increase in VAS scores with age. Figure 1B similarly shows a weak positive correlation (R = 0.2758) between years of service and VAS scores, suggesting a slightly stronger association compared to age. Both figures underscore the limited strength of these relationships. Additionally, no statistically significant association was found between BMI and VAS scores (R = 0.0011).

Figure 1 Correlation between the VAS scale variable and the variables age and length of service of midwives.

Scatter plot Fig. 2A shows the relationship between age and NDI scores, with a weak positive correlation (R = 0.1731). Scatter plot Fig. 2B illustrates the relationship between years of service and NDI scores, showing a weak positive correlation (R = 0.1987). Scatter plot Fig. 2C presents the relationship between body mass index (BMI) and NDI scores, with a weak positive correlation (R = 0.1685). The higher the age, BMI, and work experience, the higher the NDI score.

Figure 2 Correlation between the NDI scale variable and the variables age, length of service and BMI of midwives.

Figures 3A–3C presents a correlation for lumbar back pain (ODI). For example, Fig. 3A shows a moderate positive correlation (R = 0.3338) between age and ODI scores. Figure 3B illustrates a stronger positive correlation (R = 0.4074) between years of service and ODI scores. Figure 3C presents a scatter plot showing a weak positive correlation (R = 0.2718) between BMI and ODI scores. Together, the plots highlight different strengths of positive associations between ODI scores and the analyzed variables.

Figure 3 Correlation between the ODI scale variable and the variables age, length of service and BMI of midwives.

Table 2 presents the correlation analysis between various variables and three health-related indices: VAS, NDI and ODI. Statistically significant correlations (p < 0.05) are highlighted in red. There is a significant moderate positive correlation between age and both VAS (0.2476) and ODI (0.3338), as well as a weak positive correlation between age and NDI (0.1731). For BMI, there is a minimal and non-significant correlation with VAS (0.0011), but a weak positive correlation with NDI (0.1685) and a moderate positive correlation with ODI (0.2718), both of which are significant. This suggests that age, BMI, and length of service are significantly positively correlated with VAS, NDI, and ODI. On the other hand, no significant correlations are observed between the average number of working hours per month and any of the indices. The strongest correlation in the table is between the length of service and ODI (0.4074), indicating a notable association.

Table 2 Correlation between variables.

 	VAS (n = 208 )	NDI (n = 206)	ODI (n = 206)	
Age (years)	0.2476	0.1731	0.3338	
BMI (kg/m2)	0.0011	0.1685	0.2718	
Length of service (years)	0.2758	0.1987	0.4074	
Average number of working hours per month (h)	0.0198	−0.0122	−0.1454	
Notes.

Statistically significant results (p < 0.05) are highlighted in bold.

Table 3 presents the significance levels for selected relationships between work-related factors, physical activity, and three health-related indices: VAS, NDI, and ODI. The variables analyzed include the system of work, sick leave, use of painkillers, type and duration of physical activity, and frequency of physical activity. Statistically significant results (p < 0.05) are highlighted, with specific statistical tests indicated (Kruskal-Wallis or UMW). Key findings include significant differences for sick leave and painkiller use across all indices (VAS, NDI, and ODI), with higher scores observed among those who used painkillers or had taken sick leave. The type of physical activity performed also showed a significant relationship with VAS (p = 0.0054), while physical activity frequency correlated significantly with NDI (p = 0.0132). Additionally, physical activity time exhibited significant differences for NDI and ODI, particularly for longer durations of activity. These results emphasize the impact of work patterns, pain management strategies, and physical activity on the indices of pain and disability.

Table 3 Significance levels for selected relationships.

 		VAS (n = 208)	NDI (n = 206)	ODI (n = 206)	
System of work  Mean (SD)	24-hour on-call service	4.00 (0.00)	13.00 (0.00)	10.00 (0.00)	
Mixed system	5.04 (1.95)	6.64 (6.32)	8.28 (5.70)	
Single-shift system	4.85 (1.37)	7.85 (4.89)	8.74 (5.92)	
Two-shift system	4.56 (1.50)	7.79 (6.36)	7.20 (4.64)	
	p = 0.3656a	p = 0.2466a	p = 0.4351a	
Sick leave	Yes 	5.77 (1.72)	11.33 (6.44)	11.71 (4.05)	
No	4.10 (1.05)	5.96 (5.07)	5.61 (4.06)	
	p < 0.0001b	p < 0.0001b	p < 0.0001b	
Painkillers	Yes	5.07 (1.52)	8.71 (6.66)	8.96 (5.14)	
No	3.64 (1.05)	5.36 (3.64)	4.46 (2.66)	
	p < 0.0001 b	p = 0.0009 b	p < 0.0001 b	
Type of physical activity performed (intensity) Mean (SD)	Mild	4.27 (1.19)	7.66 (6.31)	7.52 (5.04)	
Moderate	5.01 (1.72)	7.91 (6.27)	8.29 (5.10)	
Intensive	4.00 (1.73)	7.00 (3.46)	4.67 (1.15)	
Not performed	4.00 (1.73)	7.56 (4.77)	5.33 (3.40)	
	p = 0.0054a	p = 0.9919a	p = 0.0865a	
Physical activity time (min) Mean (SD)	10–30′	4.82 (1.86)	9.20 (6.74)	9.18 (4.06)	
30–50′	4.58 (1.29)	6.09 (5.22)	6.31 (5.37)	
50′ and over	4.51 (1.01)	8.09 (5.99)	7.89 (5.85)	
Not applicable	4.47 (1.43)	6.26 (4.46)	4.84 (3.18)	
	p = 0.9907a	p = 0.0323a	p < 0.0001a	
Frequency of physical activity (frequency/week)	1	4.31 (1.63)	9.19 (4.80)	7.05 (5.42)	
2–3	4.80 (1.66)	8.15 (6.99)	7.93 (5.19)	
4–5	4.43 (1.32)	5.13 (4.33)	7.85 (5.04)	
6–7	4.94 (0.87)	9.17 (4.97)	7.11 (2.72)	
	p = 0.1995a	p = 0.0132a	p = 0.7011a	
Notes.

a Kruskal–Wallis.

b UMW.

Statistically significant results (p < 0.05) are highlighted in bold.

Figure 4 illustrates the relationship between VAS scores and three variables: use of sick leave, use of painkillers, and type of physical activity. Panel A shows that individuals who took sick leave reported significantly higher VAS scores compared to those who did not (p < 0.0001). Similarly, Panel B reveals that those who used painkillers had higher VAS scores than those who did not (p < 0.0001). In Panel C, the type of physical activity significantly influenced VAS scores (p = 0.0054), with mild activity associated with lower scores compared to moderate, intense, or no physical activity. These results, visualized through box plots, highlight significant differences across groups, suggesting that VAS scores, likely reflecting pain or discomfort, are affected by these factors.

Figure 4 VAS scale vs use of sick leave, (A) painkillers use (B) and type of physical activity (C).

Figures 5A–5D underscore significant associations between NDI scores and factors such as sick leave, painkiller usage, activity time, and activity frequency. Figure 5A presents a box plot comparing NDI scores between individuals who took sick leave and those who did not. The group utilizing sick leave demonstrated significantly higher median NDI scores (p < 0.0001), highlighting a strong association between elevated NDI scores and the likelihood of taking sick leave. Figure 5B illustrates the distribution of NDI scores among individuals who use painkillers versus those who do not. Participants using painkillers exhibited significantly higher NDI scores (p = 0.0009), suggesting a clear link between painkiller usage and increased disability. Figure 5C examines the relationship between activity time and NDI scores, revealing significant variation across different activity time categories (p = 0.0323). Generally, longer activity durations were associated with higher NDI scores. Figure 5D explores the relationship between activity frequency and NDI scores, showing significant differences across frequency categories (p = 0.0132). Notably, individuals engaging in 30 to 50 min of weekly activity had the lowest NDI scores, while those exercising only 10 to 30 min per week reported the highest scores. Frequency also played a crucial role: participants exercising 4–5 times per week recorded lower NDI scores than those exercising just once weekly. However, individuals who reported exercising 6–7 times weekly displayed NDI scores similar to those exercising only once a week. This suggests that excessive exercise intensity may also contribute to heightened cervical spine pain.

Figure 5 NDI scale vs use of sick leave (A) painkillers use (B) activity time (C) and activity frequency (D).

Figure 6A presents a box plot comparing ODI scores between midwives who used sick leave and those who did not. Higher median ODI scores were observed in the group that used sick leave (p < 0.0001), indicating that individuals with higher ODI scores are more likely to take sick leave. Figure 6B illustrates ODI scores for individuals who use painkillers compared to those who do not. Similar to the sick leave group, individuals using painkillers had significantly higher ODI scores (p < 0.0001). Figure 6C shows a box plot depicting the relationship between activity time and ODI scores. ODI scores varied significantly across the activity time categories, with longer activity times generally associated with higher ODI scores (p < 0.0001). These plots highlight significant differences in ODI scores based on the examined factors. Type of physical activity did not affect ODI (p = 0.0865). In summary, midwives who claimed to be physically active for more than 30 min obtained higher ODI values that were statistically significant than those who exercised between 10 and 30 min per week.

Figure 6 ODI scale vs. use of sick leave, (A) painkillers use (B) activity time (C).

Discussion

The study revealed a high prevalence of NP (97.3%) and LBP (94.6%) in the analysed group of midwives in the northern part of Poland. In a study conducted in Switzerland, it was found that among senior-year students in medical professions (midwives, nurses, physiotherapists, nutrition sciences, and occupational therapy), midwives most frequently reported lower back pain as well as neck and shoulder pain (Bucher et al., 2023). Research conducted in Iran showed that 96.7% of midwives reported at least one case of musculoskeletal disorders in the past 12 months, with the most common areas affected being the neck (45.1%) and lower back (42.9%) (Hasheminejad, Amirmahani & Tahernejad, 2023). In a study conducted in the United Kingdom, the prevalence of musculoskeletal disorders among midwives was 92%, with the lower back being the most frequently reported area (71%), followed by the neck (45%) and shoulders (45%) (Okuyucu et al., 2019). This pain is likely associated with abnormal and non-ergonomic working postures and is referred to as non-specific back pain (Weiss & Werkmann, 2009). Pain usually appears quite late in relation to the onset of development of non-ergonomic posture. This is likely to depend on the individual’s adaptive capacity. In adults, the relationship between pain complaints and work ergonomics is also quite important and should be considered from two aspects. On the one hand, the cause may be an abnormal body configuration during work activities. On the other hand, if even the body system was correct, and repeated work activities are performed repeatedly in a non-ergonomic position, this will lead to an abnormal posture over time. Both of these causes can lead to localised overload and ultimately result in the development of pain syndromes (Govaerts et al., 2021). Spinal pain is promoted, not only by the frequent stereotypical performance of various occupational activities in unnatural (non-ergonomic) positions, but also by the increasingly common sedentary lifestyle and reduced physical activity (Govaerts et al., 2021). In this situation, the onset of pain is usually only a matter of time and the effects of such a condition are referred to in the literature as work -related musculoskeletal disorders (WRMD) (George et al., 2021; Govaerts et al., 2021). Nor do they bypass those in the medical profession. This is exemplified by reports on the prevalence of WRMD among nurses, paramedics, manual therapy, occupational therapy or physiotherapy (Bryndal et al., 2023; Bryndal, Glowinski & Grochulska, 2022; Glowinski, Bryndal & Grochulska, 2021; George et al., 2021; Govaerts et al., 2021; Okuyucu et al., 2019). Spinal pain complaints and the timing of their onset may be due to the more complex substrate of the pain present and the varying compensatory capacities of the musculoskeletal system of individual subjects. Despite the usually standard work activities, the spatial arrangement of the spinal column of individuals varies, both in terms of location and body planes with the least favourable spinal arrangement. This is primarily due to the significant number of degrees of freedom of the biomechanism of the spine, and probably also to individual stereotypes and movement habits (Nowotny-Czupryna, Nowotny & Brzek, 2003; Edwards, 2005). The spinal column’s multisegmental and multifaceted mobility creates favourable conditions for compensatory displacement of body segments securing its equilibrium. However, these movements lead to overloads whose location is difficult to predict. The source of pain, however, is the lesions located at the site of accumulation of these overloads, which does not necessarily correspond to the location of spinal misalignments in working positions (Mroczek et al., 2020; Bryndal et al., 2023; Bryndal, Glowinski & Grochulska, 2022; Glowinski, Bryndal & Grochulska, 2021). Ergonomic injuries can result from a mismatch between the demands of professional tasks and the workers’ ability to meet them, especially in cases of overloading. The increased prevalence of musculoskeletal pain may be linked to staff shortages, inadequate training, extended working hours, pressure, and stress. These factors prevent nurses and midwives from focusing on proper body mechanics or taking breaks during their workday (Hochhauser & Liberman, 2024). It is essential to enhance knowledge about the causes and prevention of back pain, proper ergonomics, and the promotion of awareness of its importance in clinical practice during the education of nurses and midwives and at the early stages of their careers (Bernardes et al., 2022).

LBP is recognised in developed countries as a common cause of morbidity in a variety of occupational situations, particularly in healthcare workers, including midwives (Eurofound et al., 2017). In 2020, there were more than 500 million prevalent cases of sacroiliac pain worldwide, accounting for 7.7% of all years lived with disability and thus making up the largest share of the global disability burden. The total number of cases of sacroiliac pain worldwide is expected to rise to more than 800 million by 2050. Most of the increase in incidence will be due to population growth and an ageing population (GBD, 2023). Such a high incidence of sacroiliac pain observed in all regions around the world may carry significant social and economic consequences, especially given the high cost of care for this condition, such as expensive surgical procedures (Yelin, Weinstein & King, 2016) and the increase in opioid use (Holliday, Hayes & Dunlop, 2013). Opioid use is known to cause high rates of addiction, accidental overdose and death (Deyo, Von Korff & Duhrkoop, 2015). This generates additional costs for the individual and society in terms of medical care due to opioid misuse and loss of productivity (Florence et al., 2016). According to the GBD, there has been a moderate decrease in the age-standardised incidence rate of years lived with disability due to sacral pain compared to 1990. The authors conclude that this may be due to a change in physical work or faster recovery. Despite this, sacroiliac pain continues to be a major cause of disability worldwide, and global strategies to reduce the number of new episodes of sacroiliac pain and associated disability are crucial. There is little scientific evidence to support the effectiveness of these strategies. These should be targeted at those that are affordable and appropriate for low- and middle-income countries (GBD, 2023).

In our study, a statistically significant increase in back pain intensity (VAS), NDI and ODI index was observed in correlation with increasing age and seniority. The results of the GBD study, also confirm this for the general population and show that the prevalence of sacral pain increases with age. Similar conclusions were reached in their study by Hasheminejad, Amirmahani & Tahernejad (2023) in which they showed a significant association between age, height, BMI and musculoskeletal disorders in midwives. Older people, compared to younger adults, are more likely to be severely disabled with loss of mobility and independence resulting from sacral pain. This phenomenon leads to greater care needs (Dionne, Dunn & Croft, 2006). Both the total disability burden and disease-related costs are expected to continue to rise in the coming decades (Hartvigsen et al., 2018). Of the 21 GBD regions, the highest age-standardised prevalence of sacroiliac pain per 100 000 people was found in Central Europe (12 800; 11 500-14 400) (GBD, 2023). In 2020, 619 million people worldwide will suffer from low back pain (LBP), and it is estimated that the number of cases will increase to 843 million cases by 2050, mainly due to population expansion and ageing (GBD, 2023).

In the study presented here, respondents most frequently reported as activities generating the occurrence of pain when bending (62.14%), after lifting a load (52.43%), when standing (37.86%), when twisting the trunk (33.01%) and when sitting (31.55%). According to the GBD, nearly a quarter of years lived with disability resulting from sacroiliac pain are due to ergonomic factors at work. These factors include, as in our study, prolonged sitting or standing, bending or lifting. Sacroiliac pain is the leading cause of absence from work compared to other chronic conditions (Schofield et al., 2008). An increased risk of sacral pain is associated with occupational exposure to lifting, bending, awkward postures, vibration and tasks considered physically demanding (Roffey et al., 2010; Wai et al., 2010). In midwifery work, the greatest exposure to these factors occurs during the lifting of the mother/woman and during breastfeeding training (Hasheminejad, Amirmahani & Tahernejad, 2023). Worldwide, 11.5% of years lived with disability caused by sacral pain were attributed to such a lifestyle factor as increased BMI. Similarly, our study showed a statistically significant association of NDI (p = 0.1685) and ODI (p = 0.2718) with BMI. Obesity (Elgaeva et al., 2020; Hasheminejad, Amirmahani & Tahernejad, 2023) has been shown to be associated with the occurrence of sacral pain and the development of persistent sacral pain. however, the exact causal mechanisms of these associations remain uncertain. There is also a lack of evidence on the effectiveness of prevention strategies targeting this risk factor (GBD, 2023). In 2020 38.8% (95% UI 28.7–47.0) of global years of healthy life lost due to disability (YLD) due to back pain can be attributed to exposure to three modifiable GBD risk factors. Globally, regardless of age and gender, 22.0% (20.4–23.4) of YLD can be attributed to work ergonomic factors, while 12.5% (3.1–21.5) to smoking and 11.5% (1.4–20.9) to high BMI. The risk attributed to ergonomic work factors was highest among younger (i.e., 15–49 years) adult men (34.3%; 31.9-36- 6) and lowest among women aged 70 years and older (4.9%; 3.8−6.0). However, the risk of sacral pain associated with high BMI was highest among women aged 50–69 years (14.5%; 1.8–26.2) and lowest among younger (i.e., 15–49 years) men (9.8%; 1.2–17.5) (GBD, 2023).

It is important to consider some of the limitations of this study but also its strengths. Firstly, there was the possibility that the outcome was biased, using questionnaires containing a subjective assessment of perceived complaints, without any clinical examination to confirm the presence of symptoms and exclude overstated self-reported musculoskeletal complaints. In favour of this study is the use of clinimetric tools, which were standardised and had a high sensitivity and accuracy rate.

The size of the group was limited to only one region of Poland; to confirm the results, future research should be extended to a larger region in order to verify the findings.

This study highlights the prevalence and impact of back pain symptoms and the potentially harmful impact on professional life and patient care. The results presented in this thesis can serve as information to increase risk awareness among midwives and promote national actions to manage the risk of back pain at work. It is important to implement appropriate intervention strategies to improve ergonomic practices of midwives such as ergonomic training programs, implementing regular training sessions to educate midwives on correct posture, lifting techniques and body mechanics to prevent back pain (Bucher et al., 2023). Another important aspect is the modification of the workspace, recommending the use of ergonomic equipment such as adjustable chairs, lifting aids and appropriate workstation configurations to reduce the burden during patient care and other tasks requiring physical effort (Hochhauser & Liberman, 2024; Bernardes et al., 2022). A thorough assessment of working conditions can help to assess the likelihood of midwives developing musculoskeletal disorders and then design appropriate ergonomic solutions in the workplace. As part of workplace policies, encourage policies that promote regular breaks, job rotation and task sharing to reduce repetitive strain and prolonged periods of static postures associated with back pain (GBD, 2023; Bernardes et al., 2022). This is particularly important given the increasing average age of midwives, but also the significant shortage of midwifery staff. An important aspect is also the introduction of physical activity programmes, such as the inclusion of physiotherapy or wellness programmes aimed at strengthening the core muscles and improving overall physical fitness, reducing susceptibility to back pain (George et al., 2021). Furthermore, the importance of policy interventions at the institutional level should be highlighted, such as incorporating these recommendations into healthcare facility standards, as well as advocating for changes in midwifery education to include ergonomics as a key element of training (Bernardes et al., 2022; Chen et al., 2021; George et al., 2021).

Conclusions

This survey of midwives in Poland shows a very high prevalence of back pain among midwives. These symptoms result in mild to moderate disability according to the NDI and ODI index, sickness absenteeism and reduced activity norms, the use of pain medication for back pain and partial limitation of motor activity. Age, seniority, body mass index and low levels of physical activity are contributing factors.

Supplemental Information

Supplemental Information 1 Raw data

The results of the questionnaires from all midwives participating in the study. These results were used for statistical analysis to analyze the occurrence of back pain in the group of midwives.

Additional Information and Declarations

Competing Interests

Author Contributions

Human Ethics

Data Availability

The authors declare there are no competing interests.

Aleksandra Bryndal conceived and designed the experiments, performed the experiments, analyzed the data, prepared figures and/or tables, authored or reviewed drafts of the article, and approved the final draft.

Sebastian Glowinski conceived and designed the experiments, performed the experiments, analyzed the data, prepared figures and/or tables, authored or reviewed drafts of the article, and approved the final draft.

Kazimiera Hebel performed the experiments, authored or reviewed drafts of the article, and approved the final draft.

Agnieszka Grochulska conceived and designed the experiments, authored or reviewed drafts of the article, and approved the final draft.

The following information was supplied relating to ethical approvals (i.e., approving body and any reference numbers):

This study was approved by the Bioethics Committee at the District Medical Chambers in Gdansk (KB-14/20).

The following information was supplied regarding data availability:

The raw measurements are available in the Supplementary Files.

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
