# Peer review of "Back pain in the midwifery profession in northern Poland"

_PeerJ, doi:10.7717/peerj.19079_

## Round 0.1 · original submission · Major Revisions

Our apologies for a delayed decision. Please address the comments of the reviewers in appropriate revision

·

Basic reporting

The article is well-written in clear and professional English, providing a comprehensive background with sufficient references to relevant literature, ensuring the study's context is well-grounded in the field. The structure adheres to professional standards, with clearly defined sections, appropriate use of figures and tables, and detailed statistical analyses supporting the hypotheses. However, figure captions should be expanded to offer more descriptive explanations, making the visuals independently interpretable. While the article is self-contained and adequately addresses its hypotheses on the prevalence and impact of back pain in midwives, readability could be improved by simplifying complex statistical explanations for a broader audience. The discussion acknowledges limitations, such as the reliance on self-reported data and the study’s regional scope, but further elaboration on how these limitations affect generalizability would enhance the analysis. Including more recent literature on ergonomic interventions and prevention strategies for back pain in healthcare professionals would strengthen the practical implications of the findings. Despite these minor issues, the article meets basic reporting standards and provides valuable insights into a critical occupational health issue.

Experimental design

The experimental design is appropriate and aligns with the journal's aims and scope, addressing a relevant and meaningful research question regarding the prevalence and impact of back pain in midwives, a population often overlooked in occupational health studies. The study effectively identifies and fills a knowledge gap by examining correlations between back pain indices (VAS, NDI, ODI) and demographic or occupational factors such as age, BMI, and work experience. The investigation was conducted rigorously and adhered to high technical and ethical standards, including approval by a bioethics committee and informed consent from participants. The cross-sectional design and use of validated tools like the Neck Disability Index (NDI) and Oswestry Disability Index (ODI) enhance the study's reliability. However, while the methods section provides sufficient detail to ensure replicability, a more explicit description of recruitment strategies, potential sampling biases, and how missing data were handled would further strengthen the methodological transparency. Overall, the experimental design is robust and supports the validity of the study's findings.

Validity of the findings

The findings of the study are valid, with robust statistical analyses that are appropriately designed and controlled, ensuring reliability. The data are comprehensive and well-documented, using standardized and validated instruments such as the NDI and ODI to measure disability caused by back pain, adding credibility to the results. The conclusions are well-stated and align closely with the research questions and supporting data, clearly highlighting the prevalence, contributing factors, and impact of back pain among midwives. However, while the study provides valuable insights, its impact and novelty could be better articulated by comparing the findings more explicitly to existing global data and emphasizing the unique aspects of this population or geographic focus. Encouraging replication of the study in broader or more diverse populations would enhance the generalizability and benefit the literature. Additionally, more detailed analysis on practical interventions or policy implications would strengthen the overall contribution to the field. Despite these minor gaps, the findings are meaningful and presented with appropriate limitations.

Additional comments

- The study is limited to a single region in Poland, which may restrict the generalizability of the findings to broader populations or geographic areas.
- There is a lack of detailed recommendations for ergonomic or workplace interventions to address the identified issues, reducing the practical applicability of the research.
- Reliance on self-reported data without clinical validation introduces the risk of bias, potentially affecting the accuracy of reported symptoms and findings.
- The recruitment process and potential biases in participant selection are not clearly detailed, limiting the transparency and reproducibility of the methodology.
- Figures and tables, while comprehensive, lack sufficiently detailed captions or explanations, making them less accessible for readers to interpret independently.
- The study does not clearly articulate its novelty or how it substantially adds to existing literature beyond its focus on midwives in a specific region.
- The paper does not adequately explore actionable intervention strategies or policy implications that could help mitigate the identified occupational health risks for midwives.

·

Basic reporting

Basic reporting: Clear and professional English has been used throughout the article. Some minor mistakes have been marked for correction.
Relevant literature references along with the background of the study has been provided and established well.

Experimental design

Experimental design: Research satisfies the aims and scope of this journal.
Methods described with sufficient details.
Purpose of the research is well defined and justifies the gap and how it may fill it.
Rigorous investigation performed to a high methodological & ethical standard.
All methods are intricately described,with technical detail and sufficient information.

Validity of the findings

All fundamental data have been provided; they are statistically valid, & controlled. Conclusions are linked to original research question.

Additional comments

Introduction: Well written focusing on the previous studies on this same issue and the reason behind conducting the present investigation on the specific group of subjects. Adequate amount of background with valid references were provided. There were some minor grammatical errors which is marked in the document.

Aim: The purpose of the investigation was reasonable as particularly the specific tools haven’t been implemented before to asses the occurrence of this particular problem.

Material and methods: Ethical clearance approved. Participants consented for the study and they were informed and educate regarding the purpose of the study. Clear and logical selection criteria provided. Methodology was described with adequate details and information with proper reasoning.
Results: The relevant data has been provided and the tables regarding the statistical analysis are also presents which confirms the credibility of the investigation.

Discussion: The discussion was very well written along with literature references supporting and comparing the findings and giving a broad idea of the societal impact of this study. The limitations were highlighted and the supporting or opposing logics were provided. The reasoning seemed rational as per my review. However, there was one point which I would like to inform here. In the workers in any profession attain acclimatization during their course of work after a certain period. The influence of acclimatization to pain along with work experience would have provided additional insight to the study.

Conclusion: Briefly provided all the relevant findings, cause and effects of the investigation.

Reviewer 3 ·

Basic reporting

In the text, more emphasis is placed on presenting statistics and information, with less attention given to analyzing the relationship between risk factors and outcomes. Specifically, many similar concepts and results (such as the impact of unnatural body positions on the back) are repeated multiple times without a thorough analysis of how they relate to factors such as long working hours, poor ergonomics, or job stress. To improve this section, it is recommended to focus more on analyzing the results with an emphasis on the causal relationship between risk factors and health issues, while eliminating repetitive content and presenting the information in a more concise and conceptually clear manner.

Experimental design

1- It would have been better to consider work experience as one of the main criteria, particularly a minimum of 2 years of experience, as musculoskeletal injuries are generally less evident in individuals with less than two years of work experience, especially at younger ages.

2- It would have been better to also mention the validity and reliability of the questionnaires used.

Validity of the findings

1- If the results are clearly and comprehensively presented in a table, there is no need to repeat them in text form. In such cases, it is sufficient to refer to the table and provide a brief and concise explanation of its results. This helps the reader easily review the results and avoids redundancy.

2- If the figures represent linear regression analysis, it would have been necessary to provide the regression line equation as well. The regression equation is a crucial part of reporting regression analysis, and without it, the analysis appears incomplete. Providing the regression equation allows the reader to understand the exact relationship between the variables and even use it to predict new values.

3- The analysis and presentation of the results in this manner may lack sufficient novelty from a scientific and structural perspective. The analyses and results would have been better if they had been organized into clearer categories, such as grouping by age, work experience, or body mass index (BMI). These groupings could have helped identify more complex patterns of disorders and work-related pain. Additionally, the analysis should have more precisely examined the relationship between individual characteristics, such as age, work experience, and BMI, with the severity of pain and various disorders. These groupings could have provided more accurate and useful results for health policy and prevention of occupational disorders in different age and work experience groups of midwives. Instead of presenting results in a general way, it would have been better to provide this information through multivariate analyses and categorized presentations, allowing for more complex and precise relationships between variables to be explored, leading to richer and more practical findings.

Additional comments

Given the mentioned points, it is recommended to revise and rewrite the introduction and analysis of the results to better highlight the novelty of the study.
Thank you

Reviewer 4 ·

Basic reporting

The operational definition of back pain used for the current study is missing as inclusion of both neck pain and back pain in the study and reporting it is back pain is confusing.

Experimental design

No account of how sample size was calculated is made.
Biopsychosocial Model is a well known model for back pain but no account of physical and psychological factors at work has been made. Reporting disability and not including physical and psychological factors is worrisome.

Validity of the findings

No comment

Additional comments

The manuscript should be checked for grammatical errors in writing
Referencing should also be reviewed

Reviewer 5 ·

Basic reporting

1. Some citations are too old. Please include the latest references, at least 10 years old. English editing is required, as there are too many grammar mistakes.

Experimental design

Introduction:
1. The intro appeared to be too long, with scattered information. It isn't easy to follow the flow of points that the author is trying to convey. Too many statistical details were presented. The elements related to the originality of the article are weak. Each paragraph should discuss 1 point and elaborate on them. Remember to maintain a clear and concise writing style, using appropriate terminology and avoiding excessive jargon.

Method
1. In selection criteria, have you included those who are active in the sense of physically fit and overweight/ obese midwives, as this may influence the NDI and ODI? Poor fitness levels can reduce the body's ability to handle physical stress and overweight/ obese individuals experience greater stress on the lower back and neck.
2. Since you are including all those above 18 years old, how would you determine whether the older midwives are able to read and use an online platform? Your participants are as old as 67 years old.
3. Are there any minimum working hours to be included in the study, as those working for fewer hours might fill in the NDI and ODI differently than those with longer working hours?
4. We do not consider collecting demographic data as an interview questionnaire. Please remove the ‘interview questionnaire’.
5. Line 148: What does it mean by ‘asked about the time dimension’?
6. Line 149- 152: They were asked about the prevalence of current and past complaints of back pain. Is this part of the demographic data or ODI?
7. Test procedure: From what I understand, the NDI and ODI questionnaire was distributed online for them to answer independently. If so, this can be called an interview questionnaire, as you did not interview the participants.

Validity of the findings

Results
1. May consider reporting the demographic data in a table form.
2. Standardise the decimal points to 2 decimal points.
3. I find it difficult to follow the results as there are too many details added. Include a brief narrative summary of the key findings in the text before presenting the tables to improve the readability and flow of the results section.

Discussion
1. How do you conclude that ‘This pain can result from abnormal and non-ergonomic working posture’, when no Objective posture assessments are done? It is better to state that non-ergonomic working postures *may contribute to* or are *likely linked with* neck and back pain based on NDI and ODI findings.
2. The paragraphs appeared too long, making them difficult to follow. Please discuss one point in each paragraph for easier understanding.
3. Relate the findings to the broader literature. Discuss how this study contributes to the existing body of knowledge.

---

## Round 0.2 · accepted · Accept

Dear Dr. Bryndal,

I am writing to inform you that your manuscript - Back pain in the midwifery profession in northern Poland - has been Accepted for publication.

Congratulations!

·

Basic reporting

The article is well-structured and written in clear, professional English, but some lengthy and complex sentences could be revised for better readability. The literature review is thorough, though incorporating more globally diverse studies would enhance its relevance. Tables and figures are effective but should include clearer captions, such as defining acronyms like ODI and NDI directly. While the article addresses its hypotheses, expanding on practical applications, such as specific ergonomic interventions for midwives, would strengthen its impact. Including raw data or a detailed explanation of data handling would also improve transparency.

Experimental design

The experimental design is well-executed and aligns with the journal's Aims and Scope. The research question is clearly defined, addressing a relevant knowledge gap. The investigation meets high technical and ethical standards, with appropriate approvals and informed consent obtained. Methods are described in detail, allowing for replication. No further comments are needed.

Validity of the findings

The findings are valid, with robust and statistically sound data supporting the conclusions. The study's impact and novelty are evident, addressing a significant gap in the literature on back pain in midwives. The conclusions are well-stated, clearly linked to the research question, and appropriately limited to the results. No further comments are needed.

Additional comments

nil